# Direct observation of twisted stacking domains in the van der Waals magnet CrI₃

Myeongjin Jang [1,2,14], Sol Lee [1,2,14], Fernando Cantos-Prieto[3], Ivona Košić[3], Yue Li [4], Arthur R. C. McCray [4,5], Min-Hyoung Jung[6], Jun-Yeong Yoon[1,2], Loukya Boddapati [7], Francis Leonard Deepak [7], Hu Young Jeong [8,9], Charudatta M. Phatak [4,10], Elton J. G. Santos [11,12,13] ✉, Efrén Navarro-Moratalla [3] ✉ & Kwanpyo Kim [1,2] ✉

Van der Waals (vdW) stacking is a powerful technique to achieve desired properties in condensed matter systems through layer-by-layer crystal engineering. A remarkable example is the control over the twist angle between artificially-stacked vdW crystals, enabling the realization of unconventional phenomena in moiré structures ranging from superconductivity to strongly correlated magnetism. Here, we report the appearance of unusual 120° twisted faults in vdW magnet CrI₃ crystals. In exfoliated samples, we observe vertical twisted domains with a thickness below 10 nm. The size and distribution of twisted domains strongly depend on the sample preparation methods, with as-synthesized unexfoliated samples showing tenfold thicker domains than exfoliated samples. Cooling induces changes in the relative populations among different twisting domains, rather than the previously assumed structural phase transition to the rhombohedral stacking. The stacking disorder induced by sample fabrication processes may explain the unresolved thickness-dependent magnetic coupling observed in CrI₃.

The stacking configurations in van der Waals (vdW) crystals have a profound effect on various material properties[1]. The previous studies have demonstrated that stacking engineering is a powerful technique to achieve desired properties within the paradigm of layer-by-layer crystal engineering. Notably, the control over the twist angle between artificially-stacked two-dimensional (2D) materials has enabled the realization of non-intrinsic unconventional phenomena in moiré structures, from superconductivity up to strongly correlated magnetism[2–4].

The relative position and orientation of neighboring layers dictate the magnetic properties of CrI₃, a recently identified 2D vdW magnet[5–11]. Previous studies have shown that an interlayer stacking with monoclinic symmetry resulted in an interleaved (A-type) anti-ferromagnetic ordering in the system whereas a rhombohedral inter-layer stacking stabilizes a ferromagnetic ground state[12,13]. The thermodynamics of the two stacking orders in bulk CrI₃ result in a crossover temperature below which the layers slide from the mono-clinic to the rhombohedral ordering[14]. However, the striking

[1]Department of Physics, Yonsei University, Seoul, Republic of Korea. [2]Center for Nanomedicine, Institute for Basic Science (IBS), Seoul, Republic of Korea. [3]Instituto de Ciencia Molecular, Universitat de València, Paterna, Spain. [4]Materials Science Division, Argonne National Laboratory, Lemont, IL, USA. [5]Applied Physics Program, Northwestern University, Evanston, IL, USA. [6]Department of Energy Science, Sungkyunkwan University (SKKU), Suwon, Republic of Korea. [7]Nanostructured Materials Group, International Iberian Nanotechnology Laboratory, Braga, Portugal. [8]Graduate School of Semiconductor Materials and Devices Engineering, Ulsan National Institute of Science and Technology, Ulsan, Republic of Korea. [9]UNIST Central Research Facilities, Ulsan National Institute of Science and Technology, Ulsan, Republic of Korea. [10]Department of Materials Science and Engineering, Northwestern University, Evanston, Illinois 60208, USA. [11]Institute for Condensed Matter Physics and Complex Systems, School of Physics and Astronomy, The University of Edinburgh, Edinburgh, UK. [12]Higgs Centre for Theoretical Physics, The University of Edinburgh, Edinburgh, UK. [13]Donostia International Physics Center (DIPC), Donostia-San Sebastián, Spain. [14]These authors contributed equally: Myeongjin Jang, Sol Lee. ✉e-mail: esantos@ed.ac.uk; efren.navarro@uv.es; kpkim@yonsei.ac.kr

observation of A-type antiferromagnetism in few-layer $CrI_3$ at low temperatures compared to the ferromagnetic bulk[5,6] provided a strong indication of a layer dependence of the structural energetics in the system, where there is a clear suppression of the stacking crossover in the few-layer limit[7,15]. The spontaneous A-type magnetism onsetting at low temperatures in atomically thin $CrI_3$ has unlocked the use of atomically-thin magnets in spintronic[16,17], magnetoelectric[18,19], and optoelectronic devices[20], but most remarkably, it has also provided intrinsically-magnetic building blocks for the realization of new spin textures and magnetic ground states by controlling the stacking twist angle in their van der Waals heterostructures[21-25].

Although the stacking in $CrI_3$ was initially assumed to be either monoclinic at high temperatures or rhombohedral below 120 K[14], more recent observations have challenged this simple scenario[26]. Firstly, the layer sliding in this system appears to be incomplete even at the lowest temperatures available[26-28]. Secondly, this structural transition strongly depends on the number of layers in the crystal that has an unclear origin and dictates two thickness regimes with an unknown crossover point that appears to lie at the mesoscale[29-31]. Thirdly, the currently established monoclinic and rhombohedral structural models fail to fully explain recent temperature-dependent X-ray diffraction data[26]. Considering that layered materials often exhibit stacking faults[32], the anomalous breadth and temperature dependence of some Bragg peaks hints at a non-trivial coexistence of metastable domains of the different stacking orders with the presence of a remarkable disorder in the crystals[27]. However, conventional scattering and spectroscopic probes fail to capture the local nuances of the material, calling for new methods to unravel the structural puzzle of $CrI_3$.

Transmission electron microscopy (TEM) provides a more direct technique for the identification and quantification of defects in crystals. Yet, TEM investigation of a layered sample from the standard out-of-layer direction provides incomplete information in terms of stacking faults and their distribution[33], and a cross-sectional approach arises as necessary[34-37]. Here, we utilize plan-view and cross-sectional TEM imaging to unambiguously identify the interlayer stacking configuration in $CrI_3$ from room temperature down to 12 K. From TEM observations along the [010] crystal zone axis, we confirm the monoclinic stacking configuration of $CrI_3$ with occasional rhombohedral-type (R-type) stacking faults at room temperature. When the seemingly single-crystalline $CrI_3$ is observed along the [100] zone axis, we identify plenty of stacking domains with 120° twisted stacking faults. We systematically study the frequency of the observed different stacking faults and domain sizes, concluding that 120° twisted stacking faults account for most of the disorder in analyzed $CrI_3$. Moreover, the investigation of twisted domain distributions from exfoliated samples and as-grown unexfoliated samples reveals that the mechanical exfoliation process induces the extra twisted stacking faults in $CrI_3$. At low temperature, the monoclinic $CrI_3$ shows a change in the relative populations of twisting domains without hints of the phase transition to the rhombohedral stacking. The direct observation of various stacking disorders and their dependence on sample preparation provide key information to understand the unusual layer-dependent stacking order and magnetic properties of 2D $CrI_3$ magnets.

## Results and discussion
### Confirmation of monoclinic stacking with occasional faults
In the room-temperature monoclinic phase of $CrI_3$, the interlayer stacking is shifted by 1/3 of the monolayer lattice parameter along the zigzag lattice direction, by $(1/3)\vec{a}_1$ shift[14], as shown in Supplementary Fig. 1. On the other hand, the low-temperature rhombohedral phase shows the sliding along the armchair lattice direction, by $(1/3)\vec{a}_1 + (2/3)\vec{a}_2$ shift (Supplementary Fig. 1)[14]. Therefore, the structural phase transition can be understood by different interlayer stacking configurations in layered $CrI_3$. We first prepared plan-view TEM samples to observe the $CrI_3$ structure from the out-of-layer

direction. We paid particular attention on the preparation of pristine $CrI_3$ samples with minimized sample degradation associated with ambient exposure[38]. To this end, we used graphite flakes to encapsulate exfoliated relatively thin (thickness of 20–40 nm) $CrI_3$ flake from top and bottom and prepared plan-view TEM samples on a TEM grid inside a glovebox (Supplementary Fig. 2). The observed selected area electron diffraction (SAED) is consistent with neither single-phase monoclinic nor rhombohedral phases (Supplementary Fig. 2), which strongly suggests that $CrI_3$ samples possess unconsidered structural varieties, potentially associated with previously unidentified interlayer stacking configurations.

The unexpected results from plan-view observations warrant investigation of the interlayer stacking configuration in detail from side-view by cross-section TEM observations. The detailed preparation process for cross-sectional $CrI_3$ samples can be found in the Methods section and Supplementary Fig. 3. The cross-sectional samples appear to have negligible degradation during the FIB preparation and transfer process, as shown in Supplementary Fig. 3c and 3f. The cross-sectional $CrI_3$ samples were observed at [010] zone axis, from which we can distinguish the stacking configurations between monoclinic and rhombohedral phases (Supplementary Fig. 1). Room-temperature experimental SAED of $CrI_3$ shows an oblique diffraction pattern (Fig. 1b), which agrees well with the expected room-temperature monoclinic phase (Fig. 1c). In contrast, the rhombohedral phase of $CrI_3$ should show the rectangular SAED pattern as shown in Fig. 1d, which is inconsistent with our experimental data. The demonstrated SAED data also confirms that the vertical encapsulation with top and bottom graphite and cross-sectional TEM sampling maintains the intrinsic structure of $CrI_3$.

We performed atomic-resolution STEM imaging to study the stacking configurations at individual layer resolution. Figure 1e shows the exemplary high-angle annular dark-field scanning transmission electron microscopy (HAADF-STEM) image of $CrI_3$ from [010] zone axis. The layered structure in $CrI_3$ was clearly visualized, and the expected 1/3 shift in monoclinic stacking configuration is confirmed. The Fast Fourier transform (FFT) of the STEM image (inset of Fig. 1e) is also consistent with the observed SAED pattern at [010] zone axis. Energy dispersive X-ray (EDX) elemental mapping also clearly shows the layered Cr and I positions, consistent with the $CrI_3$ structure (Fig. 1f). We note that, due to the larger Z number and triple occupancy, the atomic columns of I appear brighter compared to Cr in the HAADF-STEM image.

Although the room-temperature monoclinic phase was confirmed, occasional stacking faults were also identified from $CrI_3$ samples. Figure 1g shows the zoomed-in electron diffraction signal, in which the vertical diffuse line is discernible. Since a diffuse line can be indicative of stacking faults, we pay close attention to HAADF-STEM images to uncover any irregular stacking configurations. As shown in Fig. 1h and Supplementary Fig. 4, we occasionally observed stacking faults with rhombohedral type (R-type) local stacking. The top-view atomic model shown in Fig. 1i displays Cr positions at an observed R-type stacking fault. Only the isolated R-type stacking faults were observed, and consecutive R-type stacking was not found in the analyzed samples. As shown in our experimental SAED (Fig. 1b), the existence of R-type faults could be easily hidden from the previous bulk structural analysis[14].

### 120° twisted stacking domains
Unexpected diffraction signals were uncovered with cross-sectional TEM imaging from a different zone axis along [100], under 30° (or 90°) in-plane rotation from the previous [010] zone axis. As shown in Fig. 2a, the experimental SAED from $CrI_3$ shows a diffraction pattern, which is not consistent with a single-crystalline monoclinic stacking configuration. Instead, SAED simulation results confirm the coexistence of twisted domains in $CrI_3$ along a vertical direction (Fig. 2b).

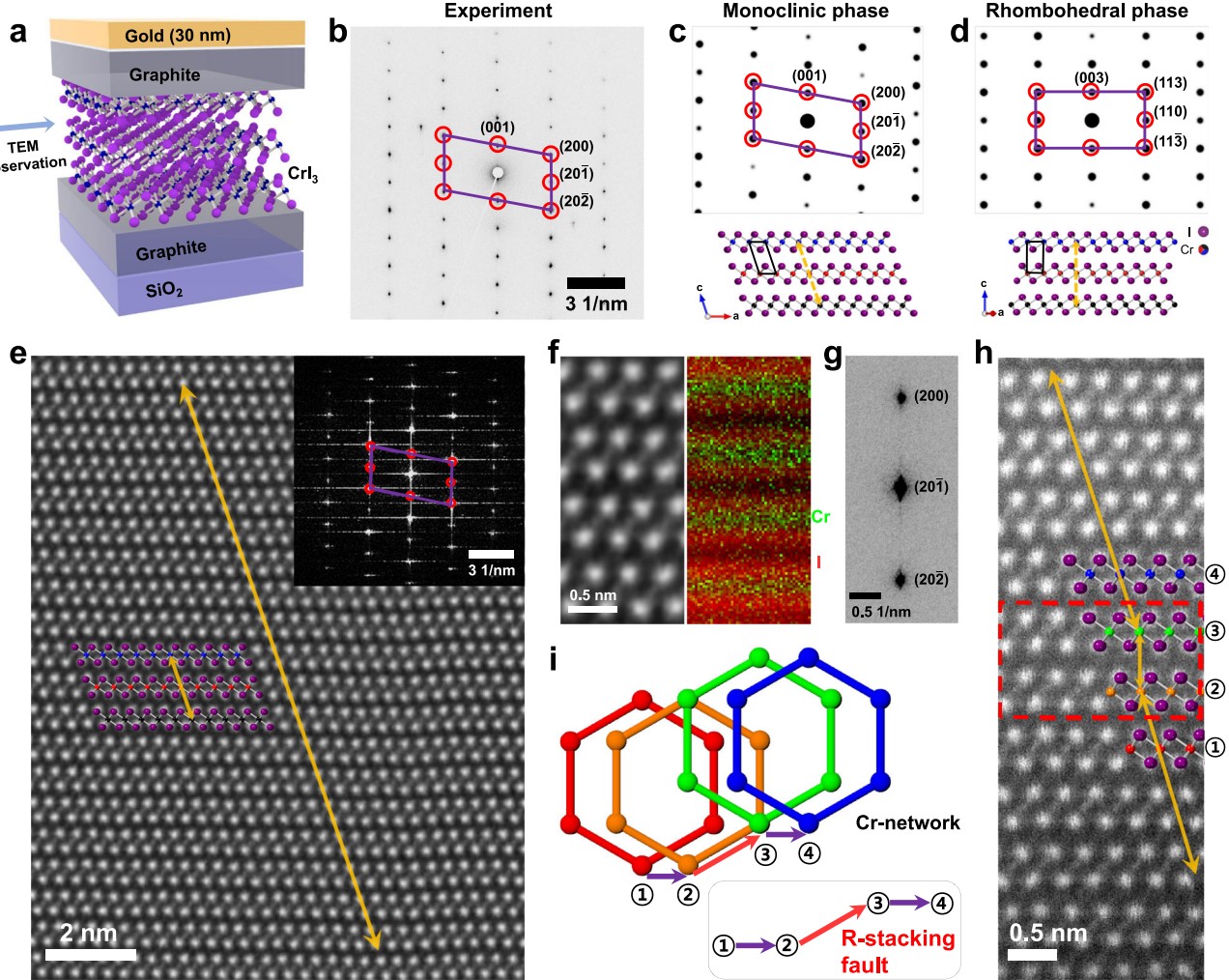

**Fig. 1 | Cross-sectional Transmission Electron Microscopy (TEM) analysis of monoclinic CrI₃ with rhombohedral-type stacking faults. a** Schematic of cross-sectional TEM samples showing the graphite-encapsulated CrI₃ crystal on a Si/SiO₂ substrate. **b** Experimental selected area electron diffraction (SAED) along [010] crystal zone axis. Some diffraction signals are labeled with a parallelogram overlay. **c, d** Simulated SAED pattern (top) and side-view (bottom) of monoclinic and rhombohedral phase stacking of CrI₃, respectively. **e** High-Angle Annular Dark-Field Scanning Transmission Electron Microscopy (HAADF-STEM) image of cross-sectional CrI₃ sample. The inset shows the fast Fourier transform of the image. The orange arrows indicate the interlayer shift with a scheme of the atomic

configuration included for clarification. **f** Zoomed-in STEM image (left) and elemental mapping (right) of Cr (green) and I (red) with Energy-Dispersive X-ray Spectroscopy (EDX). **g** Zoomed-in SAED with indexed diffraction signals. **h** Exemplary HAADF-STEM image of CrI₃ with a rhombohedral-type (R-type) stacking fault with layers labeled through 1–4 for guidance. The red dashed box indicates the R-type stacking fault, and the orientation of the arrows is used to highlight the change in stacking order throughout the layers. **i** Top-view atomic model showing the honeycomb configuration of the Cr atoms at layers 1-4 in (**h**) with an inserted R-type fault in monoclinic CrI₃. Cr atoms are colored following those at (**h**).

The diffraction signals with different colors (pink, blue, and green) correspond to the twisted variants in the CrI₃ system, in which each domain is rotated in-plane direction by 120° to give three equivalent twisted variants (Fig. 2b, d). Figure 2e shows the schematics of crystal structures by incorporating the twisted domains stacked together, where the layers with different colors indicate the twisted domains. Across the twisted stacking fault, the interlayer sliding direction is switched and breaks the symmetry along the vertical axis of the CrI₃ crystal. Due to the in-plane 120° rotational symmetry of monolayer CrI₃, 60° rotation produces a different intra-layer structure and can be distinguished from [010] zone observation. In particular, the boding direction of I-Cr-I appears with an opposite direction under a 60° rotation at [010] zone axis. The absence of such structure at [010] zone, as already discussed from data in Fig. 1, indicates that the twisted domains have the twist angle as multiples of 120°, not 60°.

Interlayer stacking configurations at 120° twisted stacking faults were investigated by atomic-resolution STEM imaging. Figure 2f shows

the HAADF-STEM image of CrI₃ along [100] zone axis (or 120° rotated from it). As shown in Supplementary Fig. 5, the rotational stacking domains can be identified as the switching of Cr directions along the out-of-plane direction from [100] zone axis. Indeed, the HAADF-STEM image clearly shows the switching of Cr directions, as shown in Fig. 2f. EDX mapping of Cr also clearly indicates the switching of Cr direction across the rotational stacking faults (Fig. 2g). The monolayer (layer with horizontal color overlay) marks the twist pivot layer along the out-of-plane direction and separates top and bottom segments (Fig. 2f, g). By incorporating information obtained from STEM images of both [010] and [100] zone axis together, we constructed an atomic model of 120° twisted domain stacking fault. Figure 2h shows the top-view of such an atomic model with a 120° rotational stacking fault. The interlayer shift direction (marked by arrows) is switched from a horizontal direction to a 120° twisted direction. The layer marked number three serves as a rotational pivot layer, which are visible from the STEM image in Fig. 2f, but not visible from [010] zone axis (Fig. 1e). We also

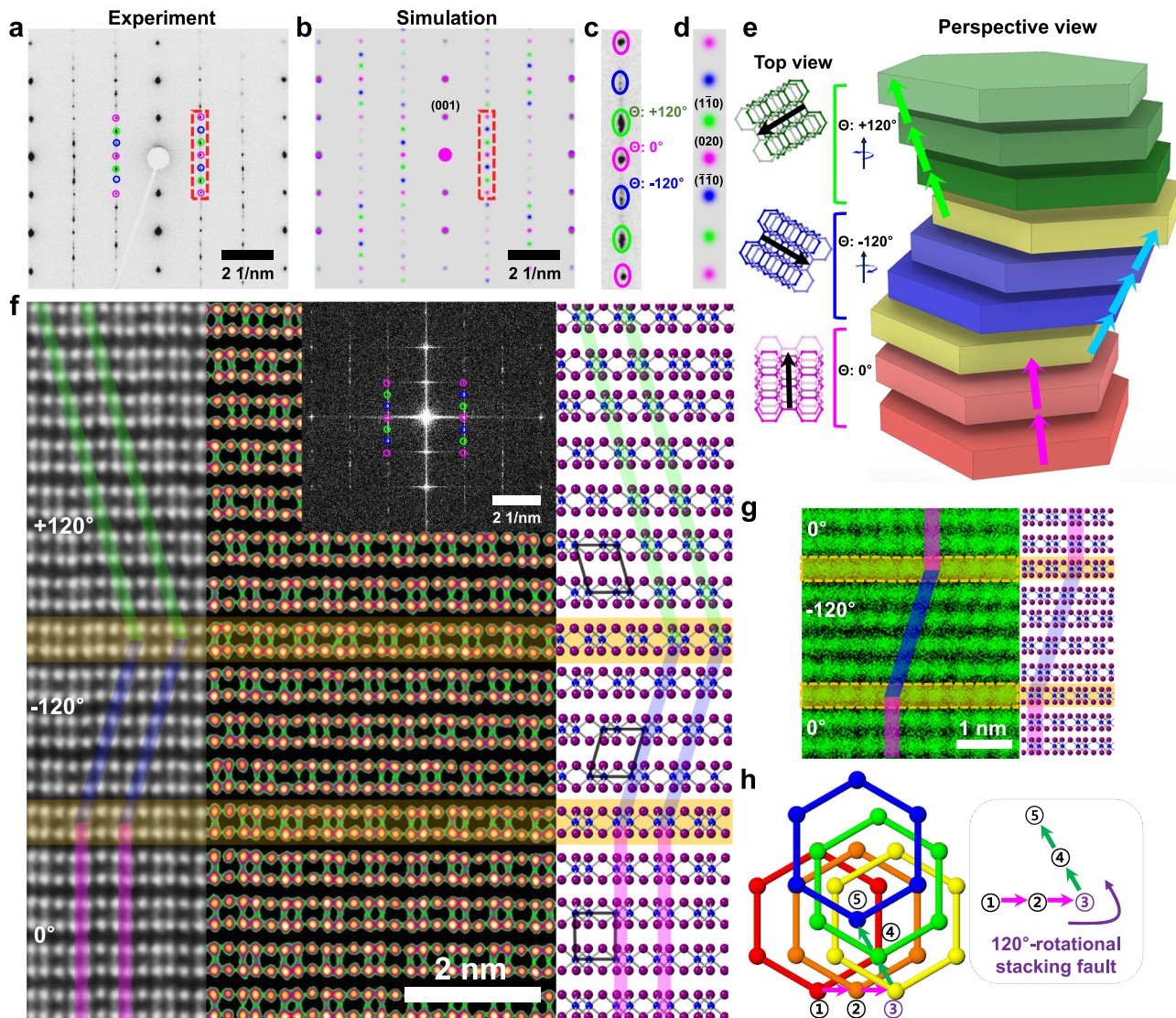

**Fig. 2 | 120° twisted stacking domains in CrI₃. a, b** Experimental and simulated SAED pattern of CrI₃ with three 120° twisted stacking domains. The signals from different domains are shown in different colors. **c, d** Zoomed-in experimental and simulated diffraction signals in the red dashed boxes of panels a and b, respectively. **e** Schematics of three 120° twisted domains vertically stacked together. The left shows the top-view atomic model for each stacking domains and the right-side schematic shows the perspective-view of twisted stacking domains. The arrows indicate the direction in which the layers are stacked, and the yellow color represents the layers where defects occur. **f** HAADF-STEM image (left) and atomic model (right) of CrI₃ with three 120° twisted domains (−120°, +120°, 0°). The locations of Cr columns are marked in color with vertical (faint purple) and diagonal (faint green and blue) lines. In the atomic model, the overlapping parallelograms and rectangles represent the unit cell. The locational of stacking faults are marked with horizontal overlay (faint yellow). The inset (top right) shows the fast Fourier transform of the image. **g** EDX Cr element mapping across 120° twisted stacking faults. **h** Top-view atomic model showing Cr positions with a 120° twisted stacking fault.

occasionally observe different types of stacking faults such as 120° twist⊕R-type as shown in Supplementary Figs. 6, 7, in which the layers after R-type-shift exhibit different interlayer sliding directions.

**Twisted domain distribution dependent on sample fabrication**
The occurrence of twisted domains and its size distribution in CrI₃ were investigated in detail. Although atomic resolution HAADF-STEM imaging is a direct way to elucidate the twisted domains and other types of stacking faults, it is not practically suitable for the analysis of large numbers of interlayer stacking shifts. Dark-field (DF) imaging can complement STEM imaging to identify and visualize twisted domains[39–41]. Figure 3b show DF images at [100] zone-axis showing different twisted domains, 0° (red) and +120° (green) domains, respectively. As shown in Fig. 3d, the distributions of twisted domain thickness identified by HAADF-STEM and DF imaging exhibit

comparable results, confirming the reliability of DF imaging for twisted domain analysis. The majority of twisted domains show the domains below 10 nm thickness for relatively thin (below 200 nm thickness) exfoliated and fully-encapsulated CrI₃ sample.

The twisted domain distributions in CrI₃ show a strong dependence on sample thickness as well as preparation methods. Figure 3e, f shows SAED and DF images from relatively thick (~430 nm) CrI₃ prepared by mechanical exfoliation and encapsulation. We found that SAED shows the diffraction signal dominated by one twisted variant. Moreover, the vertical sizes of twisted domains also display increased size compared to those of thin (<200 nm) samples. We also investigate the domain size distributions from as-grown unexfoliated samples. To maximize the preservation of the original stacking configurations from as-grown samples, FIB was directly applied to as-grown samples as shown in Supplementary Fig. 3. Figure 3g and Supplementary Fig. 9 are

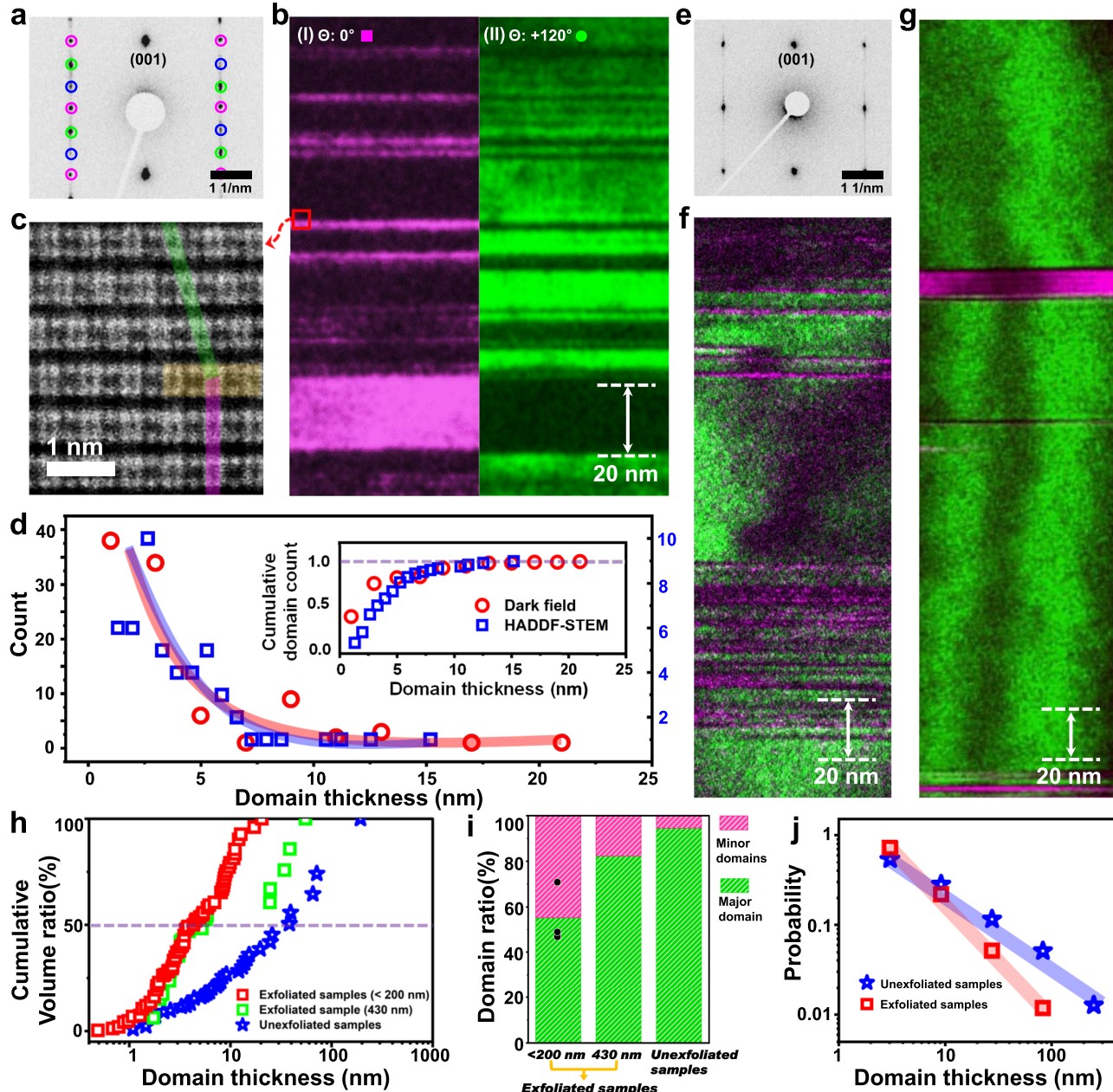

**Fig. 3 | Statistical analysis of twisted domain size according to sample thickness and preparation method. a** SAED from an exfoliated crystal of 180 nm thickness. **b** Dark-field (DF) image showing twisted domain distributions from exfoliated 180-nm-thick crystal. **c** Zoomed-in HAADF STEM image from the red box in (**b**) near a rotational domain boundary. The faint green and purple colors indicate the positions of Cr in each domain, and the yellow lines represent the stacking fault areas. **d** Comparison of domain distribution according to DF imaging (red circles) and HAADF-STEM measurements (blue squares). **e** SAED from an exfoliated crystal of 430 nm thickness. **f** DF image showing twisted domain distributions from the exfoliated 430-nm-thick crystal. **g** DF image from an unexfoliated, as-grown crystal. **h** Cumulative volume ratio by counting from smaller domain regions. **i** Relative twisted domain population from different sample preparation processes. For exfoliated samples with a thickness below 200 nm, the bar graph represents the average value, while the black circles indicate individual values from three samples. **j** Distribution of domain thickness from crystals with different sample preparation processes.

exemplary DF images of twisted domains from as-grown samples. The single-crystalline domains with bigger than 100 nm thickness are occasionally observed (Supplementary Figs. 9, 10) and the average domain size appears bigger compared to that observed from exfoliated samples. Supplementary Table 1 summarizes the sample preparation methods and thickness of samples investigated for cross-sectional TEM analysis. The as-grown unexfoliated samples show tenfold-increased twisted domain size compared to exfoliated samples, and the twisted domain structure is often dominated by one twisted variant (Fig. 3h, i).

Interestingly, we observed that the domain size in our samples follows a power law behavior as shown in Fig. 3j, in which the

distribution can be fitted by a single-slope line in the log−log plot[42]. To understand the domain distributions, we model the occurrence of stacking faults as shown in Supplementary Fig. 11. The simulation results indicate that the stacking faults in our samples are not randomly generated but display a strong correlation in their position. The bigger negative slope observed in exfoliated samples also suggests that either the overall probability of fault occurrence increases (bigger $S$) or the correlation between stacking faults decreases (smaller $k$) for exfoliated samples. To date, the mechanical exfoliation and transfer processes, which is necessary for preparation of thin samples, have been widely considered not to modify the stacking configuration in

vdW crystals. The observed stark contrast in twisted domain distribution between exfoliated and unexfoliated sample strongly indicate that the exfoliation sample preparation itself can modify the interlayer stacking configurations of vdW materials. The local bending and resulted strain of crystals during mechanical exfoliation and transfer processes can introduce the extra twisted stacking faults by interlayer shearing[43,44]. In particular, past investigations confirmed that the energy barrier for the interlayer sliding in $CrI_3$ is quite small with enhanced plasticity[28]. The observed extra stacking disorders induced by mechanical exfoliation/transfer may be also relevant in other vdW crystal systems, such as $CrCl_3$[45].

## Plan-view analysis of twisted domains

DF imaging with plan-view samples was also performed to characterize the size and distribution of the twisted domains in the lateral direction. For the plan-view DF imaging, we mainly utilized mechanically exfoliated and transferred $CrI_3$ samples as the sample fabrication directly from the bulk crystal was unsuitable. We note that the samples thicker than ~100 nm have some challenges for directly visualizing lateral domains due to the multiple-scattering during electron diffraction process and complex domain structures as the number of vertical domains grow. Supplementary Figs. 12–14 show representative DF imaging data of $CrI_3$ flakes with thickness ranging from 15 nm to 40 nm. Individual twisted domains can be identified by DF imaging by selecting relevant diffraction signals as shown in Supplementary Fig. 12d–h. The total sum of intensities in three DF images overall displays the uniform intensity, consistent with the uniform intensity observed in bright field image. We found that the lateral domain size of three twisted variants is the order of micrometers as shown in Supplementary Figs. 12 and 13. On the other hand, DF images from relatively thick samples shows the stripe pattern and the direction of domain boundaries is preferred along one direction as shown in Supplementary Fig. 14. The presence of twisted domain boundary in the lateral direction results from the local stacking transition, and the sliding direction possessing a lower energy barrier associated with stacking shift stacking will be preferred[28].

## Analysis of stacking changes at low temperatures

We explored the temperature dependence of the interlayer stacking configurations and magnetic property by performing electron diffraction and Lorentz TEM at low temperatures employing a cryoholder with as-synthesized unencapsulated samples (~2 μm thickness). Lorentz TEM imaging is a powerful tool to investigate the local magnetic property in samples[36,37,46]. Figure 4a–c are the SAED obtained at [010] zone axis from 12 K to room temperature, which confirms that the monoclinic phase persists down to 12 K with no hints of the structural phase transition to the rhombohedral stacking configuration. The absence of the structural phase transition was also confirmed by observation from out-of-plane direction (Supplementary Fig. 15). On the other hand, the analysis of the diffraction signal associated with twisting domains indicates that the relative populations among three twisting directions shows the temperature dependence. Figure 4d–i show SAED from [100] zone axis during thermal cycling between room temperature and 95 K, respectively. The population of the predominant domain labeled as Θ: 0° increases with the reduction of temperature by consuming a minor domain labeled as Θ: −120° as shown in Fig. 4j, k. Moreover, after the thermal cycling back to room temperature, the twisted domain populations did not change back to the original state. The observed hysteresis also indicates that the temperature-dependent structural change in $CrI_3$ is not simple as previously assumed and may be related to previously reported hysteresis behavior during the structural phase transition in bulk samples[14]. Temperature-dependent DF imaging of the lateral twisted domains also showed that the lateral domain population and its boundary change at low temperatures, consistent with observation of

cross-sectional samples (Supplementary Fig. 16). Lorentz TEM measurements were also undertaken with cross-sectional samples at temperature down to 12 K under various magnetic field strength (up to 2000 G) as shown in Supplementary Figs. 17 and Methods. We did not observe a noticeable Lorentz TEM contrast originating from the sample's magnetizations, which is consistent with what we expect from the observed monoclinic antiferromagnetic $CrI_3$ phase. Another 2D magnetic material, such as $Cr_2Ge_2Te_6$, exhibits the stacking-fault-insensitive magnetic domain structure[36,37,47]. The absence of various types of crystal configurations and non-stacking dependence on the magnetic properties of $Cr_2Ge_2Te_6$ is in stark contrast to the $CrI_3$ case, showcasing that the stacking configuration is indeed paramount to its magnetic ordering. Nevertheless, other compounds (CrSBr, $CrPS_4$, $MnBi_2Te_4$)[48–51] share such relationship between stacking order and magnetic coupling as that present in $CrI_3$. This indicates that a potential large family of layered compounds, either known or yet to be discovered, might show similar properties as the ones measured here. Indeed, the analysis, guidelines, and arguments included in our results might provide further pathways for additional exploration.

In summary, we utilized cross-sectional TEM analysis to uncover interlayer stacking disorders in $CrI_3$, where we found an abundance of twisted domains with 120° relative rotational stacking configuration. The mechanical exfoliation and transfer processes can introduce the extra twisted stacking faults in the crystals. For exfoliated samples, the majority of observed twisted domain sizes is smaller than 10 nm thickness, which demonstrates that the vertical domain structure should be incorporated for proper understanding of $CrI_3$ crystal structures. Conspicuous Bragg peaks found in recent X-Ray diffraction experiments[26], which could not be explained by the original monoclinic/rhombohedral structures[14], could be originated from the twisted domain texture. We believe that our observation provides key results on the understanding of the relation between interlayer stacking configuration, layer number and magnetic order in $CrI_3$ crystals. The identified twisted stacks hold promise for exploring and manipulate the unusual magnetic ground states in vdW homostructures via moiré physics as twisted layers exhibit intriguing spin correlations[8].

# Methods

## Crystal growth

High-quality $CrI_3$ crystals were grown using the chemical vapor transport technique. Chromium powder (99.5% Sigma-Aldrich) and anhydrous iodine beads (99.999% Sigma-Aldrich) were mixed in a 1:3 molar ratio in an argon atmosphere inside a glovebox. 976 mg of the mixture were loaded into a silica ampoule with a length, inner diameter and outer diameter of 500 mm, 15 mm and 18 mm, respectively. The ampoule was extracted from the glovebox with a ball valve covering the open end to prevent air exposure and subsequently, it was immediately flame-sealed and introduced into a three-zone furnace. An initial inverted gradient step was used to minimize nucleation sites in the growth zone. The source zone was raised to 650 °C, the middle growth zone to 550 °C, the third zone to 600 °C and held there for 7 days. Crystal growth takes place in center and near the cooler end of tube. After growth, crystals were extracted from the ampoule inside an Ar-filled glove box where the $O_2$ and $H_2O$ levels were maintained below 0.5 ppm to avoid material degradation. As-grown crystals were characterized by X-ray diffraction, Energy-dispersive X-ray spectroscopy and Raman spectroscopy to verify their pristine quality (see Supplementary Figs. 18–20 and Supplementary Table 2).

## TEM sample preparation

For cross-sectional TEM samples, we adapted two different methods. 1) Exfoliated thin $CrI_3$: mechanically exfoliated graphite and $CrI_3$ flakes were stacked together to fabricate graphite/$CrI_3$/graphite/$SiO_2$/Si heterostructures. The stacking process was performed with a micromanipulator under optical microscope inside an $N_2$ filled glovebox.

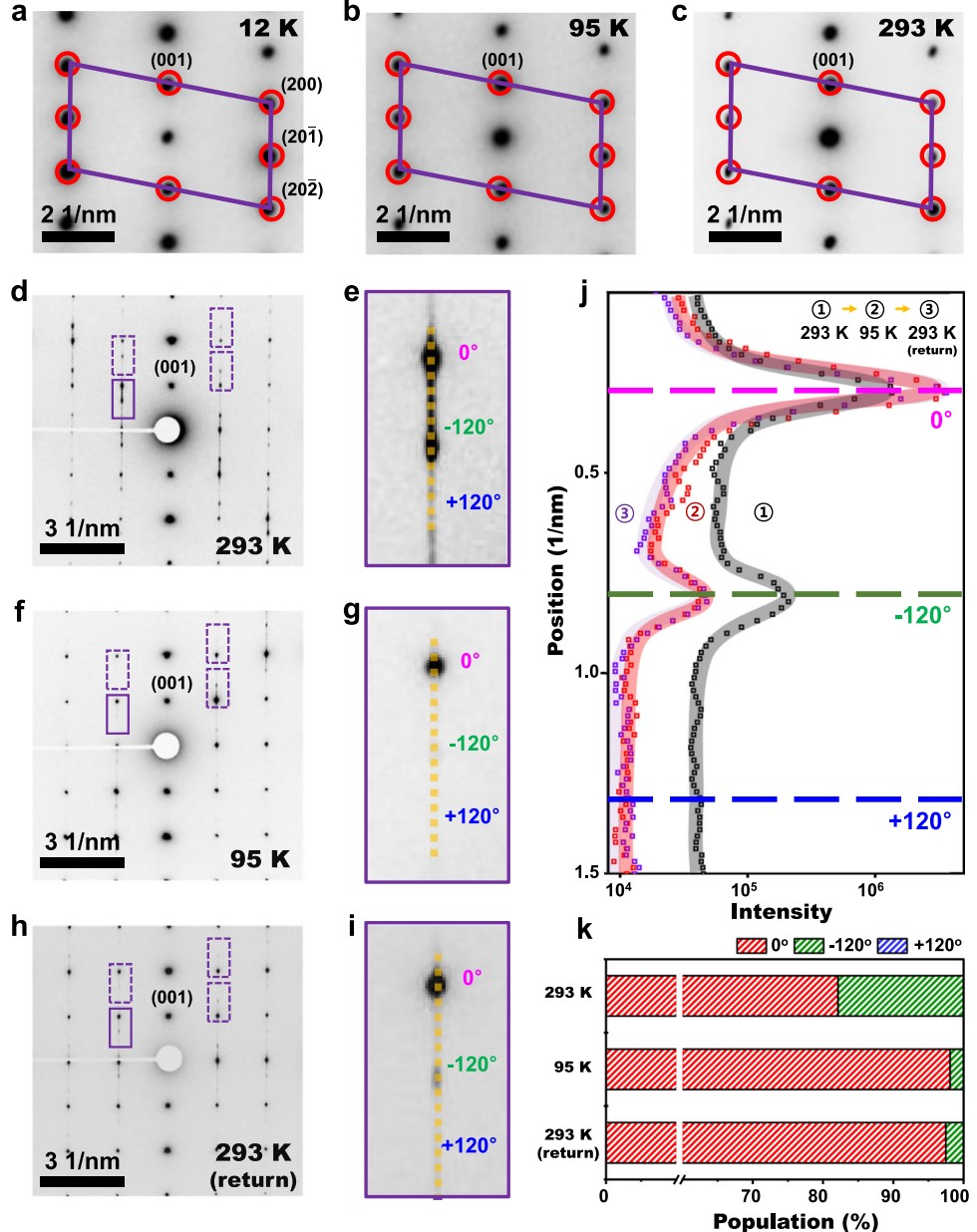

**Fig. 4 | Structural analysis of CrI₃ at low temperatures. a–c** SAED of as-grown unexfoliated CrI₃ at [010] zone axis at different temperatures, 12 K, 95 K and 293 K, respectively. **d–i** SAED of as-grown CrI₃ at [100] zone axis under thermal cycling (293 K → 95 K → 293 K) and zoomed-in SAED at the regions of rectangles, respectively. **j** Intensity analysis of SAED along the vertical yellow dashed lines in the regions of rectangles in (**d**, **f** and **h**), respectively. **k** Relative domain population among different rotational domains during thermal cycling.

Graphene was mechanically exfoliated and transferred onto a SiO₂(300 nm)/Si substrate using Scotch tape. Before transferring the CrI₃ flake, the pre-transferred graphite sample underwent annealing in a tube furnace at 600 °C for 6 hours under the flow of H₂ (30 sccm) and Ar (270 sccm) to eliminate surface residue. Subsequently, PDMS was used to stamp the exfoliated CrI₃ within a glovebox onto the previously prepared graphite/SiO₂. Employing the same method, PDMS was used to stamp the exfoliated graphene of an appropriate size onto the CrI₃/graphite, creating the graphite/CrI₃/graphite/SiO₂/Si sample structure. No polymer coating or heat treatment was applied during the dry transfer process. Lastly, the graphite/CrI₃/graphite/SiO₂/Si sample was coated with 30 nm thick Au for further protection. 2) As-grown unexfoliated CrI₃: bulk CrI₃ was coated with 200 nm thick Au for protection by thermal evaporation. The protected exfoliated or as-grown unexfoliated samples were processed by focused ion beam

(crossbeam 540, ZEISS, Ga source) for cross-sectional TEM sample fabrication. We also paid particular attention to minimize the sample transfer time from the FIB chamber to the TEM in order to avoid any contamination and/or degradation of the samples. The plan-view samples were prepared by mechanical exfoliation and transfer process onto Si₃N₄ membrane TEM grid. These samples were fabricated using a stamping method inside an N₂-filled glovebox. The samples thickness was also measured by AFM imaging inside an N₂-filled glovebox.

**TEM measurement and analysis**

HAADF-STEM imaging, SAED acquisition, DF imaging, and EDX mapping were mainly performed with a cold FEG JEM-ARM200F equipped with image and probe aberration correctors operating at 80 kV or 200 kV. The cryo-TEM experiments were carried out with a JEM-ARM 200 F by using a 626 single-tilt Gatan liquid nitrogen holder or with a

JEM-2100F by using a double-tilt Gatan liquid helium holder. Cryo Lorentz transmission electron microscopy (Lorentz TEM) experiments were performed with a JEM-2100F using Gatan liquid helium TEM holder, which has the lowest indicating temperature of 12 K. We took the in-situ Lorentz TEM images as a function of applied magnetic field and temperature to identify and isolate the magnetic phase contrast. To improve the magnetic contrast and reduce the effect of background (electrostatic phase contrast and diffraction contrast), we carried out image processing wherein, the image acquired at very high applied field of 2000G was subtracted from images acquired at lower fields. This enables for tracking the changes of magnetic phase contrast as it would respond to the changes in magnetic field.

## Data availability

The main data that support the findings of this study have been included in the main text and Supplementary Information. All other information can be obtained from the corresponding author upon request.

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

## Acknowledgements

This work was supported by the Basic Science Research Program of the National Research Foundation of Korea (NRF-2017R1A5A1014862 and NRF-2022R1A2C4002559 to K.K.) and by the Institute for Basic Science (IBS-R026-D1 to K.K.). E.J.G.S. acknowledges computational resources through CIRRUS Tier-2 HPC Service (ec131 Cirrus Project) at EPCC (http://www.cirrus.ac.uk) funded by the University of Edinburgh and EPSRC (EP/P020267/1); ARCHER UK National Supercomputing Service (http://www.archer.ac.uk) via Project d429. E.J.G.S. also acknowledges the EPSRC Open Career Fellowship (EP/T021578/1). E.N.M. acknowledges the European Research Council (ERC) under the Horizon 2020 research and innovation program (ERC StG, grant agreement No. 803092) and to the Spanish Ministerio de Ciencia e Innovación (MICINN) for financial support from the Ramon y Cajal program (Grant no. RYC2018-024736-I) and the grant PID2020-118938GA-100. This work was also supported by the Spanish Unidad de Excelencia "María de Maeztu" (CEX2019-000919-M) and is part of the Advanced Materials programme supported by MICINN with funding from European Union NextGenerationEU (PRTR-C17.I1) and by Generalitat Valenciana. F.C.P. acknowledges the MICINN for the FPU program (Grant No. FPU17/01587). Work at Argonne (to Y.L., A.R.C.M., C.M.P.) was funded by the US Department of Energy, Office of Science, Office of Basic Energy Sciences, Materials Science and Engineering Division. Use of the Center for Nanoscale Materials, an Office of Science user facility, was supported by the U.S. Department of Energy, Office of Science, Office of Basic Energy Sciences, under Contract No. DE-AC02-06CH11357. F.L.D. would like to acknowledge the funding received from the European Union, FUNLAYERS twinning project-101079184.

## Author contributions

E.J.G.S., E.N.-M., and K.K. conceived the project and guided the experiments. M.J. and S.L. mainly conducted TEM sample preparation, TEM measurements, and data analysis. F.C.-P., I.K., and E.N.-M., synthesized $CrI_3$ crystals and performed characterizations. Y.L., A.R.C.M., C.M.P., L.B., and F.L.D. performed part of low-temperature TEM experiments, Lorentz TEM experiments, and data analysis. M.-H.J., J.-Y.Y., and H.Y.J. contributed to sample preparation and TEM measurements. All authors contributed to the overall scientific interpretation and edited the manuscript.

## Competing interests

The authors declare no competing interests.
