## [Transparent Peer Review file · Nature Communications]

Direct observation of twisted stacking domains in the van der Waals magnet CrI₃

Corresponding Author: Professor Kwanpyo Kim

This manuscript has been previously reviewed at another journal. This document only contains reviewer comments, rebuttal and decision letters for versions considered at Nature Communications.

Figures originally included in the author's rebuttal have been redacted from this file.

Reviewer comments:

Reviewer #3

(Remarks to the Author)

In the manuscript "Direct observation of intrinsic twisted stacking domains in the van der Waals magnet CrI₃" resubmitted to Nature Comms, the authors have properly addressed all the questions I raised in the previous review. The additional experiments done by the authors substantially strengthened the manuscript. The finding that exfoliation introduces much more stacking faults is interesting as well. I believe the manuscript meets the standard of Nature Communications, therefore can be published in its current form.

Reviewer #4

(Remarks to the Author)

This manuscript investigates stacking domains within the monoclinic phase of CrI₃ and explores their impact on magnetic properties. Through atomic resolution electron microscopy imaging of cross-sectional samples, the authors convincingly demonstrate the presence of 120° stacking domains and elucidate their dependence on thickness and sample preparation methods. Notably, the study reveals hysteric behavior in the stacking domain population during a temperature cycle. The authors argue that no magnetic contrast in Lorentz TEM data is consistent with the absence of structural phase transitions to the rhombohedral phase at low temperatures. While the stacking domains reported here offer insights into the structural phase transition in thin layers and its implications for 2D magnetism, a clear link between these stacking domains and magnetic properties remains elusive. While the electron microscopy data, including high-quality atomic images, are carefully analyzed, the manuscript lacks essential magnetic property data. Due to this deficiency, I am unable to recommend this manuscript for publication in Nature Communications. Addressing this gap would significantly strengthen the manuscript and its contribution to the understanding of 2D magnetism.

For authors' reference, an early Lorentz TEM study on CrI₃ is reported by O Bostanjoglo and W. Vieweger "Low-temperature Lorentz microscopy on "weak" ferromagnetics", Phys. Stat. Sol. 32, 311 (1969). The thickness of CrI₃ flake in this study is not specified but a weak stripe domain contrast is clearly shown in the ab plane sample (Figure 10).

There is a Lorentz TEM study on stacking faults in another 2D magnet, Cr₂Ge₂Te₆. This study reports the absence of interaction between stacking faults and ferromagnetic interlayer coupling or anisotropy. The work shows similar atomic electron microscopy images of stacking faults (see Figs. 1f-j) and Lorentz TEM images (see Fig. 4f), both obtained from a cross-sectional sample. Drawing parallels between the findings in Cr₂Ge₂Te₆ and the current investigation on CrI₃ stacking domains may offer valuable comparative insights into the behavior of these 2D magnetic materials. The observed lack of interaction in Cr₂Ge₂Te₆ suggests that stacking faults might not significantly influence interlayer magnetic coupling.

Exploring the lateral structure of stacking domains can provide valuable insights into the effects of these domains on magnetic properties. Cross-sectional imaging might not be the most effective approach for this purpose. I suggest that the authors consider acquiring dark-field TEM images of the ab plane view. Considering the electron diffraction patterns obtained from the ab plane flake in Figure S2d, dark-field TEM images with reflections from various 120° stacking domain variants could reveal clear lateral domain distributions and sizes. This approach would allow for a more detailed analysis of how lateral variations in stacking domains are influenced by sample preparation, thickness, and temperature. Integrating

such dark-field TEM images into the study would enhance the understanding of the spatial arrangement of stacking domains and their impact on magnetic properties.

Author Rebuttal letter:

Response to Reviewers' Comments:

Manuscript #: NCOMMS-24-00144-T

Title: "Direct observation of twisted stacking domains in the van der Waals magnet CrI₃"

Author(s): Myeongjin Jang, Sol Lee, Fernando Cantos-Prieto, Ivona Kojic, Yue Li, Arthur R. C. McCray, Min-Hyoung Jung, Jun-Yeong Yoon, Loukya Boddapati, Francis Leonard Deepak, Hu Young Jeong, Charudatta M. Phatak, Elton J. G. Santos, Efrón Navarro-Moratalla, and Kwanpyo Kim

Thank you for your valuable comments and suggestions. We sincerely appreciate the time and consideration in evaluating our manuscript. Below, we have addressed each reviewer's comments to enhance our manuscript. The reviewers' comments were shown in italic, and our responses, together with changes in revision, follow.

Reviewer #3

Overall comment:

In the manuscript "Direct observation of intrinsic twisted stacking domains in the van der Waals magnet CrI₃" resubmitted to Nature Comms, the authors have properly addressed all the questions I raised in the previous review. The additional experiments done by the authors substantially strengthened the manuscript. The finding that exfoliation introduces much more stacking faults is interesting as well. I believe the manuscript meets the standard of Nature Communications, therefore can be published in its current form.

Response:

Thank you for your support. We really appreciate the reviewer's comments during the previous revision. The manuscript was significantly improved owing to the reviewer's insightful comments

Reviewer #4

Overall comment:

This manuscript investigates stacking domains within the monoclinic phase of CrI₃ and explores their impact on magnetic properties. Through atomic resolution electron microscopy imaging of cross-sectional samples, the authors convincingly demonstrate the presence of 120° stacking domains and elucidate their dependence on thickness and sample preparation methods. Notably, the study reveals hysteric behavior in the stacking domain population during a temperature cycle. The authors argue that no magnetic contrast in Lorentz TEM data is consistent with the absence of structural phase transitions to the rhombohedral phase at low temperatures. While the stacking domains reported here offer insights into the structural phase transition in thin layers and its implications for 2D magnetism, a clear link between these stacking domains and magnetic properties remains elusive. While the electron microscopy data, including high-quality atomic images, are carefully analyzed, the manuscript lacks essential magnetic property data. Due to

1

this deficiency, I am unable to recommend this manuscript for publication in Nature Communications. Addressing this gap would significantly strengthen the manuscript and its contribution to the understanding of 2D magnetism.

Response:

We appreciate the reviewer's valuable comments and suggestions. As the reviewer acknowledged, the detailed structural study (monoclinic domains persistent down to low temperature, three variants of monoclinic domains, the domain size depending on the sample preparation methods and thickness) reported in the current manuscript provides insightful information on the layered magnetic crystals. We would like to remark that the magnetic data as mentioned by the reviewer, i.e., Lorentz TEM (LTEM), was supplemented during the previous revision. It shows that a non-ferromagnetic phase is present into the samples at low temperature. Since the layers are at monoclinic phase, this is consistent with an antiferromagnetic order where no magnetic contrast can be observed in the LTEM images. Thus, this provides the missing link between structure and magnetic properties pointed out by the Reviewer. In the updated version we have made sure that this connection is clearly included.

Below we address the reviewer's specific comments to clarify some raised concerns.

Comment 1:

For authors' reference, an early Lorentz TEM study on CrI₃ is reported by O Bostanjoglo and W. Vieweger "Low-temperature Lorentz microscopy on "weak" ferromagnetics", Phys. Stat. Sol. 32, 311 (1969). The thickness of CrI₃ flake in this study is not specified but a weak stripe domain contrast is clearly shown in the ab plane sample (Figure 10).

Response: Thank you very much for pointing out the relevant reference. We agree that the mentioned reference is relevant to our manuscript. We included the reference in the revised manuscript.

Changes made:

The following reference was added in the main manuscript.

In page 11,

¶ Lorentz TEM imaging is a powerful tool to investigate the local magnetic property in samples^{35,36,45}.

References:

45. Bostanjoglo, O. & Vieweger, W. Low-temperature Lorentz microscopy on "weak" ferromagnetics. Phys. Status Solidi B Basic Res. 32, 311-321 (1969).

Comment 2:

There is a Lorentz TEM study on stacking faults in another 2D magnet, Cr₂Ge₂Te₆. This study reports the absence of interaction between stacking faults and ferromagnetic interlayer coupling or anisotropy. The work shows similar atomic electron microscopy images of stacking faults (see Figs. 1f-j) and Lorentz TEM images (see Fig. 4f), both obtained from a cross-sectional sample. Drawing parallels between the findings in Cr₂Ge₂Te₆ and the current investigation on CrI₃ stacking domains may offer valuable comparative insights into the behavior of these 2D magnetic materials. The observed lack of interaction in Cr₂Ge₂Te₆ suggests that stacking faults might not significantly influence interlayer magnetic coupling.

Response: We appreciate the Reviewer's comment. The mentioned paper presented results on the ferromagnetic properties and electrical transport behavior in Cr₂Ge₂Te₆ (CGT), another important 2D magnetic crystal. However, CGT does not show any dependence of its magnetic properties with sample thickness or number of layers (e.g., ferromagnetism/antiferromagnetism for odd/even number of layer). In this aspect, CrI₃ is quite unique on the relation between stacking order and magnetic coupling. Moreover, other vdW magnets (e.g., CrSBr, CrPS₄) more recently have been observed to show similar layer dependence on their magnetic features similarly as CrI₃. This indicates that a potential large family of layered compounds, either known or yet to be discovered, might show similar properties as the ones measured in our manuscript. Indeed, the analysis, guidelines, and arguments included in our results might provide further pathways for additional exploration for the community. We included additional discussions on other kinds of 2D magnetic crystals and comparison with CrI₃ is beneficial to our manuscript.

Changes made:

To address the Reviewer's comment, the following discussion and reference were added in the main manuscript.

In page 11-12,

¶ Another 2D magnetic material, such as Cr₂Ge₂Te₆, exhibits the stacking-fault-insensitive magnetic domain structure^{35,36,46}. The absence of various types of crystal configurations and non-stacking dependence on the magnetic properties of Cr₂Ge₂Te₆ is in stark contrast to the CrI₃ case, showcasing that the stacking configuration is indeed paramount to its magnetic ordering. Nevertheless, other compounds (CrSBr, CrPS₄, MnBi₂Te₄)^{47,48,49,50} share such relationship between stacking order and magnetic coupling as that present in CrI₃. This indicates that a potential large family of layered compounds, either known or yet to be discovered, might show similar properties as the ones measured here. Indeed, the analysis, guidelines, and arguments included in our results might provide further pathways for additional exploration.

References:

35. Han, M.-G. et al. Topological Magnetic-Spin Textures in Two-Dimensional van der Waals Cr₂Ge₂Te₆. Nano Lett. 19, 7859-7865 (2019).

36. Liu, Y. et al. Polaronic conductivity in Cr₂Ge₂Te₆ single crystals. Adv. Funct. Mater. 32, 2105111 (2022).

46. Gong, C. et al. Discovery of intrinsic ferromagnetism in two-dimensional van der Waals crystals. Nature 546, 265-269 (2017).

47. Zur, Y. et al. Magnetic imaging and domain nucleation in CrSBr down to the 2D limit. Adv.

Mater. 35, 2307195 (2023).

48. Boix-Constant, C. et al. Multistep magnetization switching in orthogonally twisted ferromagnetic
3

monolayers. Nat. Mater. 23, 212-218 (2024).

49. Peng, Y. et al. Magnetic Structure and Metamagnetic Transitions in the van der Waals
Antiferromagnet CrPS₄. Adv. Mater. 32, 2001200 (2020).

50. Deng, Y. et al. Quantum anomalous Hall effect in intrinsic magnetic topological insulator
MnBi₂Te₄. Science 367, 895-900 (2020).

Comment 3:

Exploring the lateral structure of stacking domains can provide valuable insights into the effects of these domains on magnetic properties. Cross-sectional imaging might not be the most effective approach for this purpose. I suggest that the authors consider acquiring dark-field TEM images of the ab plane view. Considering the electron diffraction patterns obtained from the ab plane flake in Figure S2d, dark-field TEM images with reflections from various 120° stacking domain variants could reveal clear lateral domain distributions and sizes. This approach would allow for a more detailed analysis of how lateral variations in stacking domains are influenced by sample preparation, thickness, and temperature. Integrating such dark-field TEM images into the study would enhance the understanding of the spatial arrangement of stacking domains and their impact on magnetic properties.

Response: Thank you for your valuable comments. As suggested by the reviewer, we have conducted extra experiments on the in-plane lateral domains in CrI₃. Specifically, 1) we have performed dark-field TEM imaging with plan-view CrI₃ samples with various thickness, and 2) observed the changes in domain structures at low-temperature (liquid N₂ temperature). The added data clearly exhibit that the lateral stacking domains are the order of micrometers. Moreover, DF images from relatively thick samples shows the stripe pattern and the direction of domain boundaries is preferred along one direction. The presence of twisted domain boundary in the lateral direction results from the local stacking transition. Therefore, the formation of domain boundaries will be strongly influenced by the sliding direction possessing a lower energy barrier associated with stacking shift, and the observed preference of the domain boundary formation can be attributed to the direction of strain applied during sample exfoliation/transfer process. In addition, we directly observed that the stacking domain changes upon cooling, which also provides extra information how the switch in stacking domain occurs. As the temperature decreased from room temperature to lower temperatures, the minor domain areas decreased while the predominant domain areas increased, which is consistent with the cross-section TEM analysis in the manuscript. We believe that the provided extra experimental data significantly improve the quality of our work, and our manuscript is ready to be accepted promptly.

Changes made:

To address the Reviewer's comment, the following discussion and Supplementary Figures S12-14 and S16 were added.

In page 10,

DF imaging with plan-view samples was also performed to characterize the size and distribution of the twisted domains in the lateral direction. For the plan-view DF imaging, we

mainly utilized mechanically exfoliated and transferred CrI₃ samples as the sample fabrication directly from the bulk crystal was unsuitable. We note that the samples thicker than ~100 nm have some challenges for directly visualizing lateral domains due to the multiple-scattering during electron diffraction process and complex domain structures as the number of vertical domains grow. Supplementary Figures S12~14 show representative DF imaging data of CrI₃ flakes with thickness ranging from 15 nm to 40 nm. Individual twisted domains can be identified by DF imaging by selecting relevant diffraction signals as shown in Supplementary Figure S12d~h. The total sum of intensities in three DF images overall displays the uniform intensity, consistent with the uniform intensity observed in bright field image. We found that the lateral domain size of three twisted variants is the order of micrometers as shown in Supplementary Figures S12 and S13. On the other hand, DF images from relatively thick samples shows the stripe pattern and the direction of domain boundaries is preferred along one direction as shown in Supplementary Figure S14. The presence of twisted domain boundary in the lateral direction results from the local stacking transition, and the sliding direction possessing a lower energy barrier associated with stacking shift stacking will be preferred²⁷.

In page 11,

Temperature-dependent DF imaging of the lateral twisted domains also showed that the lateral domain population and its boundary change at low temperatures, consistent with observation of cross-sectional samples (Supplementary Figure S16). ...

In page 15,

... The plan-view samples were prepared by mechanical exfoliation and transfer process onto Si₃N₄ membrane TEM grid. The samples thickness was measured by AFM imaging inside an N₂ filled glovebox. â

[REDACTED]

Supplementary Figure S12. Visualization of stacking domains in plan-view CrI₃ samples of 14-nm thickness using dark-field TEM imaging. (a) Optical image of a mechanically-exfoliated CrI₃ flake on Si₃N₄ membrane TEM grid (membrane thickness: 30 nm). (b) AFM topography image of the CrI₃ flake. The AFM measurement was performed inside a glovebox filled with N₂. (c) The AFM line profile along the red line in panel b. The sample thickness is approximately 14 nm. (d) Electron diffraction pattern of the sample. The signals from three monoclinic variants with different sliding directions are marked with violet circles and labeled as D1, D2, and D3. (e) Bright-field TEM image of the sample. (f-h) Dark-field TEM images obtained by selecting D1, D2, and D3 peaks. (i) A total sum image combining panels f, g, and h. (j) Intensity profiles of the DF images and their sum across the green line in panel f. The local twisted domain thickness is estimated from the intensity in the DF images. The purple dashed lines indicate the location of domain boundaries. (k) A schematic representation of a proposed lateral twisted domain structure derived from the line intensity profiles of DF images.

[REDACTED]

Supplementary Figure S13. Visualization of stacking domains in plan-view CrI₃ samples of 20-nm thickness using dark-field TEM imaging. (a) AFM topography image of the CrI₃ flake. (b) The AFM line profile along the red line in panel a. The sample thickness is approximately 20 nm. (c) Bright-field TEM image of the sample in region a. (d-f) Dark-field TEM images obtained by selecting D1, D2, and D3 peaks. The sample edge is aligned with the zigzag lattice direction as shown in panel d. (g) A total sum image combining panels d, e, and f. (h) Bright-field TEM image of the sample in region b. (i-k) Dark-field TEM image obtained by selecting D1, D2, and D3. (l) A total sum image combining panels i, j, and k.

[REDACTED]

Supplementary Figure S14. Visualization of stacking domains in plan-view CrI₃ samples of 40-nm thickness using dark-field TEM imaging. (a) AFM topography image of the CrI₃ flake. (b) The AFM line profile along the red line in panel a. The sample thickness is approximately 40 nm. (c) Bright-field TEM image of the sample. The inset is electron diffraction from the sample. (d-f) Dark-field TEM images obtained by selecting D1, D2, and D3 peaks. The sample edge is aligned with the zigzag lattice direction as shown in panel d. (g) A total sum image combining panels d, e, and f.

[REDACTED]

Supplementary Figure S16. Observation of lateral domain changes in CrI₃ at low temperatures using DF imaging. (a) Observation of lateral domain changes with temperature variation. While the D1 region remains largely unchanged with temperature, a transition from the orange dashed region of D3 to the D2 region is observed as the temperature decreases. (b) An image illustrating disappeared and emerged domains using color indicators. Red areas indicate domains that vanished as the temperature dropped from room temperature to a lower temperature, while green areas indicate newly formed domains.

Reviewer comments:

Reviewer #4

(Remarks to the Author)

The authors conducted further experiments on ab plane samples, revealing the lateral stacking domain variants and their evolution upon cooling. Ideally, magnetic imaging below T_c on the ab flakes with such stacking domains would conclusively validate the key findings of this study. Nevertheless, the additional experimental data provided sufficiently addresses my comments, and I recommend this manuscript for publication.
